# Spatial 3D Printing of Continuous Fiber-Reinforced Composite Multilayer Truss Structures with Controllable Structural Performance

**DOI:** 10.3390/polym15214333

**Published:** 2023-11-06

**Authors:** Daokang Zhang, Xiaoyong Tian, Yanli Zhou, Qingrui Wang, Wanquan Yan, Ali Akmal Zia, Lingling Wu, Dichen Li

**Affiliations:** State Key Laboratory for Manufacturing Systems Engineering, Xi’an Jiaotong University, 28 Xian Ning West Road, Xi’an 710049, China; zhangdk0821@stu.xjtu.edu.cn (D.Z.); zyl1631118746@163.com (Y.Z.); wqr@qzc.edu.cn (Q.W.); 15332080756@163.com (W.Y.); ziaaliakmal@stu.xjtu.edu.cn (A.A.Z.); lingling.wu@mail.xjtu.edu.cn (L.W.); dcli@mail.xjtu.edu.cn (D.L.)

**Keywords:** spatial 3D printing, continuous fiber, lightweight structure, mechanical properties

## Abstract

Continuous fiber-reinforced composite truss structures have broad application prospects in aerospace engineering owing to their high structural bearing efficiency and multifunctional applications. This paper presents the design and fabrication of multilayer truss structures with controlled mechanical properties based on continuous fiber-reinforced thermoplastic composite 3D printing. Continuous fiber composite pyramid trusses fabricated by 3D printing have high specific stiffness and strength, with maximum equivalent compression modulus and strength of 401.91 MPa and 30.26 MPa, respectively. Moreover, the relative density of a truss structure can be as low as 1.45%. Additionally, structural units can be extended in any direction to form a multilayer truss structure. Structural performance can be controlled by designing the parameters of each layer. This study offers a novel approach for designing a multifunctional multilayer truss structure, a structure with low-density needs and unique load-bearing effects.

## 1. Introduction

Continuous fiber-reinforced composites are used for a wide range of applications in the automotive and aerospace industries owing to their excellent mechanical properties, which include low density, high specific strength, and high modulus [1,2,3,4]. Because 3D printing is flexible and customizable, it has improved the use of continuous fiber-reinforced materials in industrial applications and made it possible to fabricate complex structures [5,6,7]. Studies on 3D printing of continuous fiber composites have primarily focused on ways of increasing fiber content and achieving a denser microstructure for better mechanical characteristics [8,9,10]. However, the interest in the design and manufacture of lighter structures with superior mechanics has been increasing among researchers.

Topological optimization [11] and lattice structure [12,13] designs are extensively utilized in 3D printing to produce lightweight designs, and the optimization method is used to produce a variety of composite structures. The topological optimization of composite structures has been studied extensively. The difficulties in the topological optimization of continuous fibers, such as the fiber discontinuity, length scale separation, decreased design freedom, and fiber orientation of CFRSs with complicated forms, have recently been resolved by a number of studies. Li et al. [14], for instance, provided a brilliant path-designed 3D printing method based on the composite’s evaluated stress states, taking into account the load transmission path and anisotropic property of the continuous fiber filament. Based on this methodology, Wang et al. [15] developed a load-dependent path planning method under the stress-vector-tracing algorithm for 3D-printed CFRPCs, where fiber trajectories are generated along the load transmission path. In order to produce these, Tian et al. [16] created a multiscale design and production approach that included the concurrent optimization of fiber orientation and macro structural topology, taking into account hatch space and printed fiber radius. Continuous fiber composite structures with simultaneously optimized fiber orientations and topology structures were produced by 3D printing. At present, regardless of the kind of topology optimization method, the path planning is carried out within a plane, and compared with the spatial structure, there is still a certain gap in the density and specific intensity. For the lattice structure design, 2D CFRSs with various filling forms [17], including rectangular, circular, honeycomb, rhombus, trapezoidal, and corrugation, can be created using the conventional layer-by-layer 3D printing technique. Unique cell shapes, such as negative Poisson’s ratio units [18] and re-entrant units [19], are also used to create 2D cellular architectures. Due to the limitations of the additive manufacturing process, such as requiring layer-by-layer stacking, it is difficult to manufacture spatial lattice structures.

Some researchers are starting to manufacture 3D lattice structures using new processes. First the continuous carbon fiber-reinforced thermosetting composite grid was fabricated with the winding printing approach [20]. Similarly, He et al. [21] printed spring and grid structures using UV-curable thermosets. Eichenhofer et al. [22] printed an ultra-lightweight sandwich structure with a pyramidal truss core using a continuous lattice fabrication process. In addition, Li et al. [23] and Luan et al. [24] manufactured continuous fiber-reinforced thermoplastic lattice structures using a free-hanging 3D printing method. However, research into space composite structures still needs to be further improved from the aspects of technology and structure. Owing to the unique properties of continuous fibers and the requirement for a continuous path, it is challenging to create continuous fiber composites with a general structure [25,26]. However, because most partial composite truss structures need to be fabricated and assembled, the overall fabrication of truss structures remains a challenge [27]. The integrated fabrication of continuous fiber composite truss structures still requires further research, and the mechanical properties of continuous fiber truss structures are still lacking, especially the control of the mechanical properties of multilayer truss structures.

In this study, multilayer pyramid truss structures with variable structural parameters are designed and fabricated using continuous fiber-reinforced thermoplastic composite additive manufacturing technology for pyramid truss structures with different structural parameters and by employing spatial path planning, as opposed to the conventional additive manufacturing process of layer-by-layer manufacturing. Pyramid trusses with different structural parameters were tested for compressive strength, and the effects of different structural parameters on the mechanical properties of the pyramid truss structure were theoretically analyzed. Based on the study of the pyramidal truss structure, horizontal and vertical extensions of the pyramidal truss were carried out, allowing for the extension of the truss unit in all directions. Each layer of the multilayer truss was individually designed to obtain a multilayer truss structure with special mechanical properties. In the face of the shortcomings of existing research, the design and manufacture of multilayer truss structures with different mechanical properties were achieved through the design of parametric truss structures on the basis of continuous fiber additive manufacturing. The manufacture of structures with better mechanical properties was completed at as low a cost as possible while still being able to cope with a variety of different load environments.

## 2. Design and Manufacturing

### 2.1. Pyramid Truss Unit Design

Truss structures are widely used in various industrial scenarios, and the concise and efficient design of a structure can be achieved by programming its mechanical properties [28]. We needed to choose a truss for parametric design, and in the end, we chose the pyramid truss because the pyramid truss has the following advantages: the structural form is simple and there are fewer structural parameters, which is conducive to using the structural parameters to control the mechanical properties; the structure is stable and uniform, with a strong bearing capacity; and the shape of the structure makes it easy to plan a continuous print path, with no overlap in the paths in the important bearing structure part, which is the most important reason. Maintaining a continuous print path can achieve better mechanical properties. As shown in Figure 1, the main parameters included in the unit are truss length (L), truss inclination angle (θ), truss width (d), truss thickness (h), and overlap thickness at the fold (Δx). A pyramid truss structure was formed by overlapping two continuous fiber composite filaments; furthermore, the four fulcrums at the bottom were connected and fixed with composite filaments. Here, the pyramidal cells are extended in two ways: (1) along the direction of the pyramidal cell composite wire, which allows for a simple array in both the horizontal and vertical directions but has the disadvantage of having many layer-to-layer individual nodes, which prevents the formation of tight connections and force transfer; (2) along the 45° direction, which doubles the relative density compared with the first method and allows for connections between the nodes and a tighter bond between the layers in the vertical direction.

The relevant parameters of the pyramid truss structural unit can be obtained using the following calculation: 

The base length of the pyramid truss structural element is expressed as follows:(1)$$L_{b}=\sqrt{2}\times \left(L\times cos\theta +∆x\right)$$

The height of the pyramid truss structural element is expressed as follows:(2)H=L×sin⁡θ+h

The relative density $\bar{\rho }$ of the pyramid truss structural element is expressed as follows:(3)$$\bar{\rho }=\frac{\left(4\times \left(L+∆x\right)+4\times L\times cos\theta \right)\times b\times h+2\times b^{2}\times h}{{L_{b}}^{2}\times H}$$

### 2.2. The Manufacture of Pyramid Truss Structure

#### 2.2.1. Material and Equipment

The pyramid truss structure was fabricated using a continuous fiber-reinforced thermoplastic composite 3D printer (Shanxi Fibertech Technology Development Co., Ltd., Xi’an, China), as shown in Figure 2. In this study, continuous fiber-reinforced thermoplastic composites were composed of continuous aramid fibers for reinforcement and polylactic acid (PLA) polymer material as the matrix. Combining the two materials can provide a composite truss with a better bearing capacity and a lighter relative density. In the present study, a continuous aramid-fiber material (Kevlar fiber with linear density of 670 dtex, density of 1440 kg/m^3^, breaking strength of 152 N, specific strength of 23.0 cN/dtex, modulus of 700 cN/dtex, from DuPont Corp) with excellent stability in 3D printing for CFRCs was used as the reinforcement material, and polylactide (PLA/1.75 mm, density of 1240 kg/m^3^) from Polymaker in China was used as the thermoplastic material.

To prevent interference between the nozzle and forming structure, a custom extended nozzle was used to finish the printing process. The outer layer of the nozzle was pasted onto a thermal insulation layer to make the temperature inside the nozzle as uniform and as constant as possible. The wire-feeding motor fed the resin wire into the heating chamber. The continuous fiber and melt resin were completely compounded inside the heating chamber. The composite monofilament was extruded from the nozzle while being pulled by the continuous fiber and rapidly cooled by an air cooler.

#### 2.2.2. Pyramid Truss Manufacturing Process

The printing of a continuous-fiber thermoplastic composite space truss differs from that of the standard 3D printing layer-by-layer manufacturing procedure. Although the printing path and forming structure can be freely designed in space, spatial printing results in the deformation of the printed portion owing to the nozzle movement, which changes the printing results. Therefore, it is necessary to replicate the print path.

The nozzle temperature analysis produced the following results: The mechanical properties of thermoplastic resin changed with the temperature; specifically, its stiffness decreased with increasing temperature. As a result, after cooling, the section that was farther away had a higher rigidity, whereas the part closer to the nozzle always maintained a lower rigidity. The material in this portion was softened and easily distorted when the nozzle moved because the temperature of the thermoplastic resin near the nozzle was lower than its glass transition temperature (Tg). The length of the softened part depended on the nozzle temperature and cooling rate and remained a fixed value during the printing process. As a result, when the nozzle moved, certain continuous traction force was generated in the fiber, which deformed the printed structure. To avoid this deformation, it was necessary to re-plan the printing path to ensure that the final printing result was consistent with the designed structure. 

In the truss 3D printing process, the main errors arose from the following three factors: First, the deformation of the structure was caused by the traction force of the nozzle on the printed structure during the printing process, as shown in Figure 3. For this part, we changed the initial print path to print a larger tilt angle and then printed the designed tilt angle by moving it horizontally. We denoted the angle deviation value and nozzle-generated traction force as ∆θ and *F*, respectively. The traction force perpendicular to the truss structure is $F_{x}$. The deflection of the truss is regarded as the process of bending the single rod of the truss, and the maximum deflection $\mu_{max}$ of the beam deformation under a simple load gives the formula for the angle deviation value ∆θ, as follows:(4)$$∆\theta =\frac{F*\sin\theta *{\left(l+∆l\right)}^{2}}{3EI}$$

For instance, we selected a pyramid structure with a truss length of 12 mm and a tilt angle of 50° for theoretical calculation. The process parameters of the 3D printer included a nozzle diameter of 1.6 mm and a nozzle temperature of 230 °C to ensure the quality of the unsupported printing, and the adopted printing speed was 50 mm/min. The tractive force F of the printing nozzle under this process parameter was measured with a tension meter. The modulus of elasticity of the material was E (the elastic modulus of the material here refers to the mechanical properties of continuous fiber-reinforced thermoplastic composites [8]), and we could easily calculate the angular deviation.

Second, in the nozzle lifting and falling process, because the nozzle diameter d was not negligible, the position of the truss node was shifted to the left by Δ = d/2 via upward lifting. Similarly, when the nozzle fell, the position of the truss node shifted to the right by Δ. The third factor focused on the effects of pressure and heat on the lower layer when printing the upper structure when connecting trusses on each floor, as well as the bending deformation of the printed structure caused by gravity when printing horizontally without support. It is necessary to ensure that the fibers are always tensioned during the printing process. At this time, the printed structure maintains a certain straightness owing to the traction force and achieves a certain stiffness requirement through rapid cooling.

This study adopted measures such as error compensation and maintaining continuous fiber tension to achieve high-precision printing. To achieve this, the print path of the structure was designed, and the corresponding difference compensation was calculated. Furthermore, the specific coordinate points printed for each path and the resin wire feed rate were calculated, which could be used to convert these coordinates into the G-code for the 3D printer using MATLAB. 

## 3. Result and Discussion

This section investigates the mechanical compression performance of continuous fiber-reinforced thermoplastic composite truss structures. Further design and manufacturing of the multilayer pyramid structures can be achieved by studying the mechanical properties of truss structures with different structural parameters. A compression rate of 1 mm/min was used to test the sample, Mechanical tests of the CFRTCTT samples were performed using a universal testing machine (MTS 850/25t, MTS Corp., New York, NY, USA).

### 3.1. Single-Layer Pyramid Structure

The pyramidal structural unit was expanded into a 2 × 2 array structure as a performance sample for compression testing, as shown in Figure 4. Furthermore, two main parameters of the continuous fiber composite pyramid truss structure, the inclination angle θ and truss size L, were studied. Here, the other parameters were constant, for example: truss width (d = 2 mm), truss thickness (h = 0.4 mm), and volumetric fiber content (8.75%) (compared with normal printing, the volume content of fiber in this sample was lower because the resin extrusion volume of the space truss was higher). In this study, we designed two groups of experiments. The first group, Group 1, had a fixed inclination angle of 50° and truss length L of the pyramid structure ranging from 6 mm to 18 mm, and each group consisted of three samples, with the same parameters for truss lengths differing by 2 mm. The second group was named Group 2, in which the length of the trusses was fixed at 12 mm and the truss inclination angle of each group of samples varied from 30° to 65° every 5°, and the number of samples in each group was also 3. The relative density of the pyramid truss with different structural parameters was calculated based on the design of the pyramid truss structure in Section 2, as shown in Table 1 and Table 2.

The compression performance of a pyramidal truss unit was directly influenced by two crucial structural variables: truss length and truss angle. Here, we plot the force–displacement curve of the pyramidal truss structure in compression under various parameters using the data from a batch of samples that best reflect this structural parameter, as shown in Figure 4. Evidently, when the tilt angle was gradually increased from 30° to 65°, the load capacity of the pyramidal truss unit increased with the increasing tilt angle, and the maximum load capacity increased from 173 N to 485 N when the length of the truss was fixed at 12 mm. Therefore, from the perspective of energy absorption, an increase in the angle can further increase the energy absorption. However, for another structural parameter, the length of the truss, which gradually increased from 6 to 18 mm, the load capacity of the pyramidal unit decreased as the length of the truss increased; furthermore, when the length of the truss was increased, the energy absorption decreased. At a fixed tilt angle of 50°, the maximum load capacity dropped from 689 N to 130 N. From the force–displacement curves, it can be concluded that these two structural parameters are important for the pyramidal truss. This result can be easily explained because as the tilt angle increased, the direction of the pressure was closer to the direction of the continuous fibers. For composite structures, the more the direction of the force agreed with the direction of the material, the better the mechanical properties that the structure could exhibit. For lattice structures, the smaller the lattice size, the better the mechanical properties, so the structure showed that the smaller the truss size, the better the load-bearing performance. Therefore, by quantitatively analyzing the influence of the two parameters on the mechanical properties, pyramidal trusses can be designed with different structural parameters for different demand scenarios.

It is not comprehensive to judge structural parameters based only on their maximum load-carrying capacity. Further analyses should be conducted by comparing their equivalent strength and equivalent stiffness and the effect of different structural parameters on the mechanical properties of the pyramid truss. The equivalent bearing capacity, equivalent strength, and equivalent stiffness of the pyramid truss structure were calculated under different parameters. Their equivalent bearing capacity, equivalent strength, and equivalent stiffness, are the results obtained from the tests divided by the relative densities of the different structures. As shown in the Table 3 and Table 4, with an initial increase in the angle, the equivalent bearing capacity of the pyramid truss with variable angles increased rapidly. When the angle was close to 60°, the specific bearing capacity had little difference, while the equivalent strength and equivalent stiffness increased significantly with the increase in the angle; furthermore, the maximum equivalent strength was 28.52 MPa, and the equivalent stiffness was up to 401.91 MPa. For a pyramid truss with a variable length, its equivalent bearing capacity exhibited a relatively complex law, and the equivalent strength and stiffness increased with a reduction in the truss length. When the truss length was L = 6 mm, the maximum equivalent strength was 30.26 MPa and the equivalent stiffness was 251.96 MPa. The mechanical behaviors of the 3D-printed pyramid truss structure also should be compared with those of the conventional truss structure, such as the pyramidal fiber composite lattice by Xiong [29] and the tetrahedral fiber composite lattice by Zhang [30]. The strength and stiffness of the pyramid truss structure created in this paper are not as good as those of the standard process truss structure in terms of mechanical parameters. In terms of equivalent strength, the truss structure in this study had a lower relative density and performed similarly to the standard process truss structure.

From the compression test results of the single-layer pyramid structure sample, we obtained the rule of the influence of these two parameters on the mechanical properties of the period. However, further analyses enabled us to design a pyramid-structured sample with controllable mechanical properties.

### 3.2. Multilayer Pyramid Structure

After systematically analyzing the relationship between the properties of the pyramidal unit structure and structural parameters, we further verified that the mechanical properties of the pyramidal unit were controllable in a multilayer structure. In the previous section, we accomplished the design and fabrication of a single layer of 2 × 2 pyramidal cells, in which it was necessary to fabricate a multilayer pyramidal structure; therefore, we designed several types of multilevel pyramid truss structures, as shown in Figure 5.

The first type of pyramid truss adopted a vertical linear array. The apex of each pyramid structure was at the center of the upper pyramid structure. The advantage of this stacking method is that it maintains a low relative density, although this involves high requirements for the printing process and path planning. For the pyramid truss of this vertical array, samples with the same and different structural parameters for each floor were designed as three-floor truss structures. The height of each floor of the former was 10 mm with a 45° truss inclination. The heights of each floor of the latter from top to bottom were 10, 8, and 6 mm, respectively. The inclination also changed with height. In the previous section, based on the influence of the structural parameters on the mechanical properties, we speculated that, for a uniform single-layer truss structure, the stress of each layer is consistent when it is compressed. Under theoretical conditions, the three-layer structure undergoes uniform deformation and eventually fails simultaneously. For a non-uniform structure, owing to the difference in the bearing capacity and compression strength of structures with different parameters, the structure fails from top to bottom.

The second type of multilayer pyramid truss structure was designed by combining the pivot of the upper pyramid element with the vertex of the lower pyramid element; thus, the number of elements in each floor is different. The number of pyramid elements on each floor from top to bottom was 1, 4, and 9, respectively. The advantage of this design is that the overall structure with overlapping nodes is more stable and reliable, and the force transmission is more continuous during the compression process. The overall structure remains a pyramid, reflecting the design of a multilevel structure. Although the structural parameters of each floor are the same, the experiment inevitably leads to failure from top to bottom, owing to the large difference in the number of units on each floor. However, designing the structural parameters can enable us to design structures with similar bearing effects between the three floors. Therefore, the original design sample had a uniform pyramid truss structure. The truss inclination of each floor was 45°, the truss length was 14 mm, and the unit structural parameters of each floor were consistent. Through calculations, we designed the truss inclinations of the bottom floors to be 7°, the truss inclination of the second floor to be 28°, and the truss inclination of the top floor to be 60°. The truss length changed with a change in inclination. Under this design, the load capacity of each floor was relatively close, and an overall slow failure could be realized.

The compression of a multilayer pyramidal truss structure was analyzed using the compression test data and photographs, as shown in Figure 6. For pyramidal truss structures arranged directly in the vertical direction, regardless of whether the structure had three layers of the same height or three layers with varying height gradients, it maintained a certain load capacity until the overall structure failed and deformed completely. Evidently, for pyramid structures designed at the same height, when the strain was 0.78, all the three layers underwent deformation. Additionally, the second layer of the structure was damaged first; then, the top layer structure was compressed; and finally, the third layer of the structure was fully compressed owing to the printing accuracy. However, for pyramid structures designed at different heights, it can be clearly seen that the top layer was compressed first. From the influence of the structural parameters on mechanical performance, it can be seen that the top layer’s parameters lead to the failure of its structure more easily thus indicating a top-down failure form.

From the test results, it is evident that, for the second type of multilayer pyramid truss structure, layer-by-layer failure occurs in compression, with collapse deformation occurring from top to bottom. The force–displacement curves both show three peaks corresponding to the peak forces at the three layers when the damage occurs. In the case of the uniform pyramidal truss structure, the form of failure of the individual cells was the same as that of the single-layer cell structure. The load capacity of the layer rapidly decreased when the structure failed and the layer compressed to its limit. The load capacity then began to increase to the peak load capacity of the next layer, which resulted in the failure of the next layer structure, and the load capacity rapidly decreased. The ratio of the three peak forces to the number of truss units in the three-layer structure was 1:4:9. For the non-uniform structure, the failure form and overall damage process were similar to those of the uniform structure. Because of the adjustment in the inclination angle of each layer, the peak forces of the three-layer structure failure were relatively close to each other, indicating that special overall mechanical properties can be achieved through the adjustment of the unit structure. In general, this proves that it is feasible to design the structural parameters of a unit structure to affect its mechanical properties.

There may be some deviation between the predicted situation and experimental results owing to the printing error and instability of the truss connection at each floor. However, through the design of the structural parameters, it was found that the mechanical behavior of the pyramid truss realized under different conditions changed significantly and was close to the designed situation. The mechanical properties of each layer can be controlled through various aspects such as the number of units in each layer and the angle of the truss. Furthermore, a pyramidal truss structure with a specific mechanical behavior was obtained, demonstrating controllable mechanical behavior. 

### 3.3. Discussion

#### 3.3.1. Theoretical Analysis

To obtain further quantitative effects of the structural parameters on the mechanical properties, the relationship between the structural parameters and the load bearing should be obtained. We conducted theoretical modeling for a pyramid truss structure with four inclined trusses. The mechanical properties of the continuous fiber composite space truss structure were systematically studied and analyzed, as shown in Figure 7. First, a mechanical analysis for a single truss unit in the truss structure was carried out: the length of the unit was set to L, the inclination angle was set to θ, the cross-sectional area was A, the variation in the height of the truss was ∆h, and the modulus of elasticity of the material was E (the elastic modulus of the material here refers to the mechanical properties of continuous fiber-reinforced thermoplastic composites [8]).

Ideally, the stress of a single truss is uniform; therefore, we only analyzed the stress of one truss. The distance of movement δ downwards was decomposed into the distance along the truss direction $\delta_{a}$ and that perpendicular to the truss direction $\delta_{s}$. Similarly, the force along the truss direction and that perpendicular to the truss direction were divided into $F_{a}$ and $F_{s}$, respectively. The relationship between the stress and the strain in the elastic deformation range by Hooke’s law can be obtained as follows:

The axial force $F_{a}$ of a single truss:(5)$$F_{a}=\frac{EA}{L}\times \delta_{a}=\frac{EA}{L}\times ∆h\times \sin\theta$$

The normal force $F_{s}$ of a single truss:(6)$$F_{s}=\frac{12EI}{L^{3}}\times \delta_{s}=\frac{12EI}{L^{3}}\times {∆h}_{}\times \cos\theta$$

The load on a single truss:(7)$$F=F_{a}\times \cos\theta +F_{s}\times sin\theta =\frac{EA\times L^{2}\times {sin}^{2}\theta +12EI\times {cos}^{2}\theta }{L^{3}}\times ∆h$$

For instance, we selected a pyramid structure with a truss length of 12 mm and a tilt angle of 50° for theoretical calculation. According to the experimental results, the displacement when the maximum peak force was reached was 0.75 mm, so we calculated that the overall bearing capacity of the truss was 337 N when the displacement was 0.75 mm through Formula (7), and we considered this theoretical calculation value to be the theoretical maximum bearing capacity of the truss under the structural parameters. From the theoretical calculation, it can be concluded that the most important physical parameters affecting its compression performance are the tilt angle θ of the truss structure and the single rod length L of the truss. These two physical parameters directly affect its mechanical performance; therefore, we studied different unit length trusses and different tilt angle trusses with the unit length L at 6 mm~18 mm and the tilt angle at 30°~65°. For different length units, although the reduction in truss length brings about greater load-carrying performance, the relative density of the truss structure further increases and the relative density of the truss structure needs to be considered simultaneously with the load-carrying performance. The theoretical calculation results and the results of the experimental data trends remained consistent, and the error was within a certain range of control. The main reason for the error could be the instability of the printing process. From the experimental data and theoretical structure, we can explain the angle and length on the mechanical properties of the structure with the specific influence. Therefore, we can design and estimate the pyramid structure with different performance outputs.

#### 3.3.2. Failure Deformation Analysis

The response of the structure to load deformation determines the ability of the pyramidal truss structure to exhibit excellent equivalent load bearing, equivalent strength, equivalent modulus, and low relative density. Evidently, when the material properties are the same, the key structures act as stresses. In the pyramid-shaped truss structural unit proposed in this paper, the bottom truss serves to fix the position of each inclined truss while the peripheral frame is created to guarantee the continuity of the path throughout the continuous fiber printing process. The peak load of the truss without an external frame was reduced by 15% compared with the truss with an external frame, as can be seen from the mechanical characteristics of the original and frameless connection samples. Furthermore, the bottom truss undoubtedly plays an important part in the overall truss’s stability during compression; the peak load of the truss without the bottom truss is exceedingly low, as illustrated in Figure 8.

Understanding the truss failure modes is crucial for developing high-performance trusses. As shown in Figure 8, when the pyramid truss structure is subjected to compression deformation, there are various deformation modes of truss bending deformation, such as fiber and resin matrix separation, truss structure fracture, and truss bending. Our failure criteria here include structural load failure (when the peak force dropped to 80%, the structural failure mode could be easily found), the various failure forms, the structural deformation buckling, the material fracture at the macro level, as well as the separation of the fiber resin interface at the micro level. The different failure modes are caused by the uneven stress of the four inclined trusses during structural compression, so the trusses in the same direction receive greater impact, resulting in more serious failure. When the four trusses are subjected to more uniform forces, all four trusses will bend in the same direction of rotation and the structure will not undergo a complete destructive failure, such as fiber and resin matrix separation, truss structure fracture, and truss bending. Among the various failure forms, the overall structural load-bearing performance is better when the four trusses constituting the pyramid structure fail uniformly at the same time.

## 4. Conclusions

In this paper, a multilayer pyramid truss structure with different structural parameters was manufactured by using the method of space continuous fiber thermoplastic composite additive manufacturing. Firstly, through the optimization of the printing path and the design of the process parameters, the manufacture of the pyramid truss structure with relatively high precision was realized. On this basis, experimental and theoretical studies were carried out on the pyramid truss with different structural parameters, the relationship between the structural parameters and the bearing performance was obtained, and the manufacture of a pyramid truss structure with controllable mechanical properties was finally realized. The preliminary results indicated that a 3D-printed continuous fiber composite truss structure is a feasible and promising load-bearing structure. Truss structures with different mechanical properties can play different roles in dealing with different bearing environments. For example, in the case of heavy loads, a truss with a higher bearing performance can realize the load of the final structure, while in the case of low loads, the truss can achieve a certain deformation and energy absorption effect. It may also be possible to achieve a shock absorption effect in the field of construction. This possibility provides a potential design method for multilayer structures in the future.

## Figures and Tables

**Figure 1 polymers-15-04333-f001:**
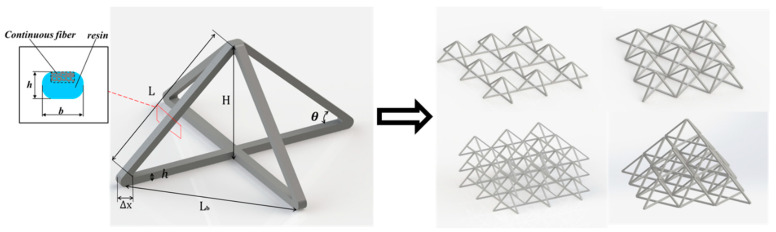
Pyramid composite truss units and their different types of arrays stacked in 0/90°and 45° directions.

**Figure 2 polymers-15-04333-f002:**
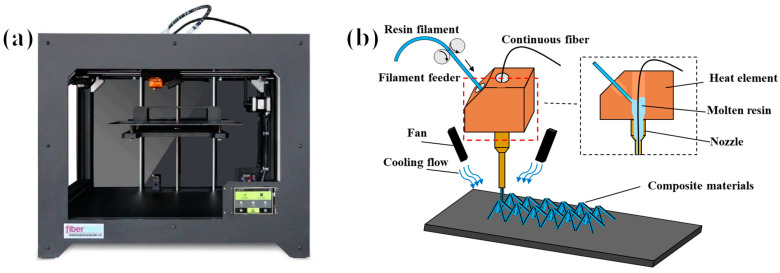
(**a**) Continuous fiber composite 3D printer physical image, and (**b**) continuous fiber composite space structure printing schematic.

**Figure 3 polymers-15-04333-f003:**
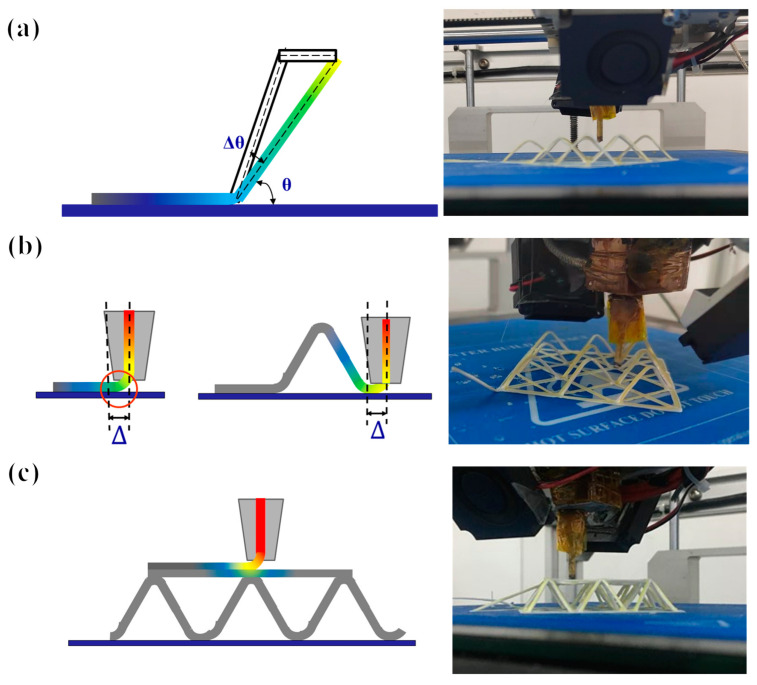
Main errors of continuous fiber-composite-printed space truss structure: (**a**) inclination angle error, (**b**) lateral error of starting and landing point, and (**c**) inter-layer error.

**Figure 4 polymers-15-04333-f004:**
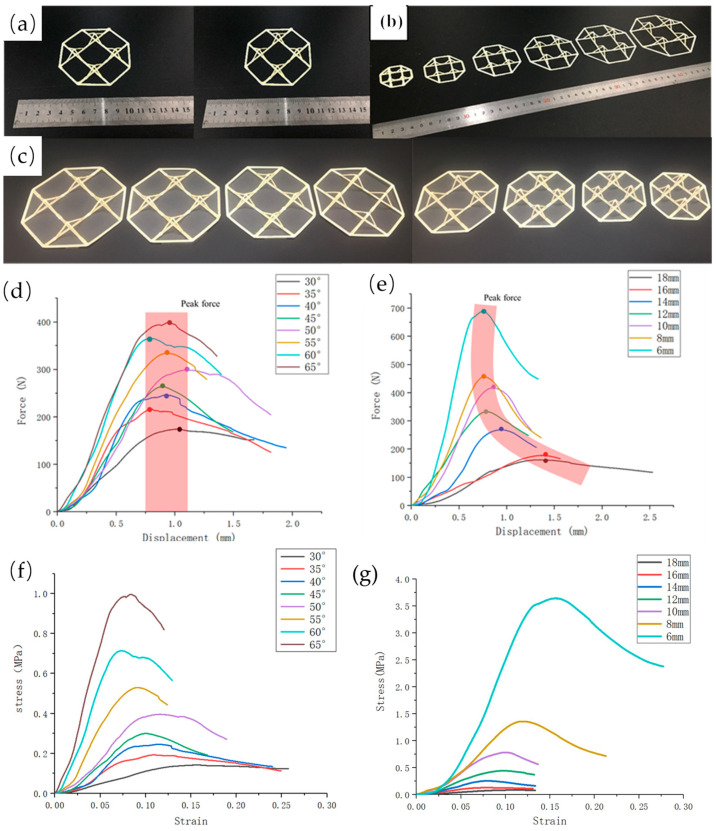
(**a**) The 2 × 2 pyramidal truss unit structure samples, (**b**) pyramidal truss structure samples of different scales, (**c**) pyramidal truss structure samples of different tilt angles, (**d**,**f**) compressive curves with different truss inclination angles, and (**g**,**e**) compressive curves with different truss scales.

**Figure 5 polymers-15-04333-f005:**
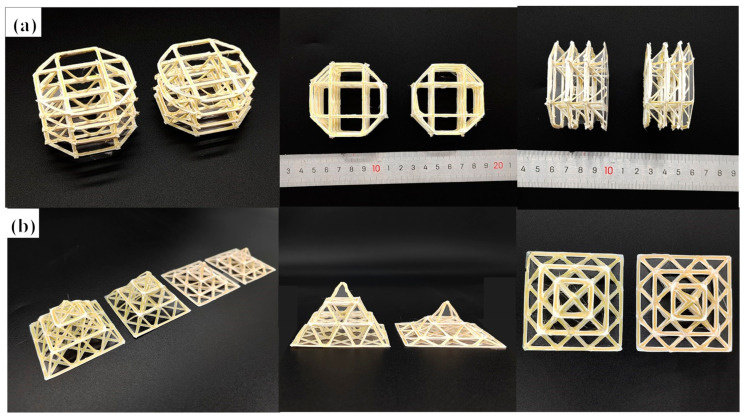
Samples of multilayer truss structure: (**a**) vertically stacked three-layer truss sample (uniform structure design and non-uniform structure design per layer), and (**b**) 45° connected stacked three-layer truss sample (uniform structure design and non-uniform structure design per layer).

**Figure 6 polymers-15-04333-f006:**
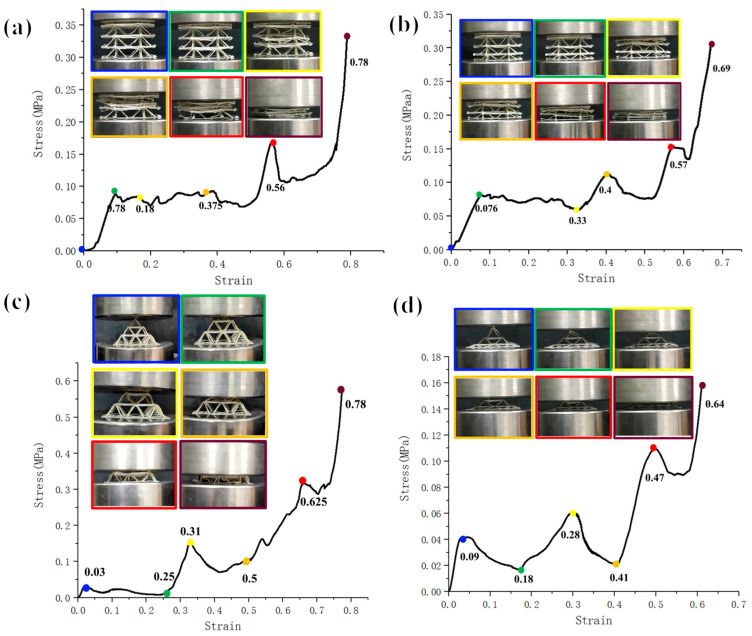
Compression performance curves of different multilayer pyramid structures. (**a**,**b**) compression performance curves of vertically stacked three-layer truss sample (uniform structure design and non-uniform structure design per layer), and (**c**,**d**) ompression performance curves of 45° connected stacked three-layer truss sample (uniform structure design and non-uniform structure design per layer).

**Figure 7 polymers-15-04333-f007:**
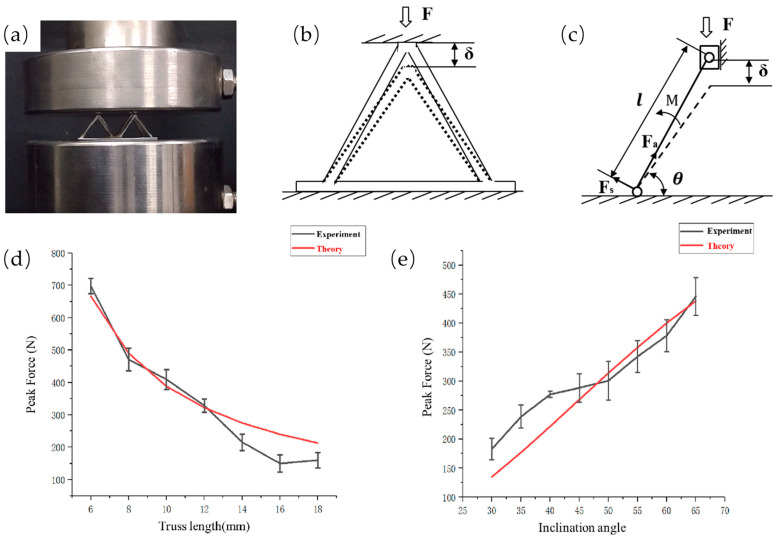
(**a**) Actual photo of 2*2 pyramid truss unit compression, (**b**) force analysis diagram of pyramid truss unit, (**c**) force analysis of simplified pyramid single truss, and comparison of experimental data and theoretical peak force: (**d**) different truss length and (**e**) different inclination.

**Figure 8 polymers-15-04333-f008:**
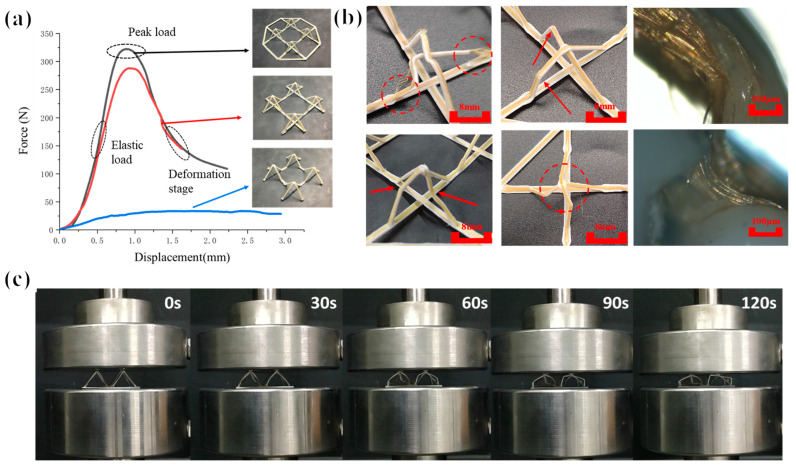
(**a**) Compression failure performance of different structures, (**b**) compression failure damage form of inclined trusses, and (**c**) deformation in different stages.

**Table 1 polymers-15-04333-t001:** Structural parameters (variable angle) and relative density.

Type	Inclination Angle θ	Truss Size L/mm	$\mathrm{Relative}\;\mathrm{Density}\;\bar{\rho }$
A1	30°	12	2.98%
B1	35°		2.81%
C1	40°		2.74%
D1	45°		2.76%
E1	50°		2.88%
F1	55°		3.11%
G1	60°		3.48%
H1	65°		4.06%

**Table 2 polymers-15-04333-t002:** Structural parameters (variable size) and relative density.

Type	Inclination Angle θ	Truss Size L/mm	$\mathrm{Relative}\;\mathrm{Density}\;\bar{\rho }$
A2	50°	6	8.30%
B2		8	5.46%
C2		10	3.86%
D2		12	2.88%
E2		14	2.23%
F2		16	1.78%
G2		18	1.45%

**Table 3 polymers-15-04333-t003:** Structural parameters (variable angle) and mechanical behaviors.

Type	$\mathrm{The}\;\mathrm{Equivalent}\;\mathrm{Maximum}\;\mathrm{force}\;F/\bar{\rho }$ (N)	Equivalent Strength σ$/\bar{\rho }$ (MPa)	$\mathrm{Equivalent}\;\mathrm{Elastic}\;\mathrm{Modulus}\;E/\bar{\rho }$ (MPa)
A1	1451.34	4.73	37.82
B1	2117.43	7.57	68.89
C1	2454.38	9.80	99.39
D1	2463.77	11.21	124.48
E1	2586.80	13.71	164.42
F1	2813.50	17.83	228.03
G1	2765.80	21.61	291.59
H1	2850.98	28.52	401.91

**Table 4 polymers-15-04333-t004:** Structural parameters (variable size) and mechanical behaviors.

Type	$\mathrm{The}\;\mathrm{Equivalent}\;\mathrm{Maximum}\;\mathrm{Force}\;F/\bar{\rho }$ (N)	Equivalent Strength σ$/\bar{\rho }$ (MPa)	$\mathrm{Equivalent}\;\mathrm{Elastic}\;\mathrm{Modulus}\;E/\bar{\rho }$ (MPa)
A2	2075.30	30.26	251.96
B2	2138.28	20.97	210.62
C2	2610.10	18.38	185.19
D2	2586.81	13.71	164.42
E2	2296.14	9.49	117.34
F2	1853.93	6.14	55.55
G2	2603.45	7.10	67.11

## Data Availability

The data is included in this article.

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
