# Peer review of "Spatial 3D Printing of Continuous Fiber-Reinforced Composite Multilayer Truss Structures with Controllable Structural Performance"

_polymers, 2023, doi:10.3390/polym15214333_

Round 1
Reviewer 1 Report
Comments and Suggestions for Authors
1. In section 1. Introduction, it is recommended to add references to articles on topological optimization and form finding of composite structures.
2. How does the formula (2-3) take into account the volume of the overlapping two continuous fiber composite filaments zone at the top and bottom of the structure?
3. Check the title of chapter 2.2. D printing of pyramid truss structure.
4. In section 2.2.1. Material and equipment, describe the properties of aramid fibers and PLA resin wire (manufacturer, density, number of fillers, mechanical properties, etc.)
5. In section 2.2.2. Pyramid truss manufacturing process, it is recommended to give an example of calculation using the formula (2-4).
6. In section 3. Result and discussion, it is necessary to specify experimental data for truss width (d), truss thickness (h). What is the volumetric fiber content in the composite?
7. Specify in formulas (3-1) – (3-3) what the symbol Δh means.
8. Give examples of calculations using the formula (3-3). Specify values for all parameters used.
9. How was the theoretical peak force determined in Figures 7d and 7e? What failure mode was described and by what failure criteria?
10. How are you planning to continue this research?
Author Response
Dear reviewer:
Thank you for your decision and constructive comments on my manuscript. We have carefully considered the suggestion of Reviewer and make some changes. We have tried our best to improve and made some changes in the manuscript.

Reviewer 2 Report
Comments and Suggestions for Authors
The paper shows a very interesting approach in the case of AM production of polymer composites with the use of continuous fibers. Unfortunately, there are some issues that need to be improved:
1. Line 16 - correct the unit "MPa".
2. Line 17 and 3.1 paragraph - are you sure density should be shown in "%"?
3. I cannot understand why you mentioned metal AM methods instead of focusing on a deeper analysis of the present state-of-the-art thermoplastics. It is especially important in the case of your introduction which suffers from a lack of a proper and specific literature review. I mean some exact examples (i.e. Smith et al. did this.... and Kowalski et al. did that).
4. At the end of your introduction I cannot see any justification of the aim of your work. Please specify it better. Why did you choose a pyramid truss - not some other lattice structures ???
5. Why have you used PLA as a matrix? This type of material is characterized by many disadvantages for some exact applications (low strain level, non-UV resistance, etc.).
6. Figures 4, and 6 - provide the charts in Stress-Strain relationship - no Force - Displacement.
7. Figure 8 - these figures should be shown as micro - no macro images.
8. All conclusions should be completely rewritten - how do you want to use PLA-based composites in aerospace and other applications? Please point out some exact outcomes of your work and use i.e. bullets.
Comments on the Quality of English Language1. There are some fundamental issues in using technical language. i.e.: "topology optimization" instead of "topological optimization".
Author Response

(The authors gave the same response as above.)

Round 2
Reviewer 2 Report
Comments and Suggestions for Authors
The authors made significant improvements, but there are some minor issues that should be corrected:
1. Still I suggest using stress - strain charts instead of force - displacement.
2. Scales in figure 8b should be more visible. I suggest putting scales also in macro images.
Author Response
Dear reviewer:
Thank you for your decision and constructive comments on my manuscript.We have further improved our paper, and modified it according to your suggestions. We believe that our paper has become more excellent. As for the new questions raised, we will respond to each one in the following paragraphs.
- Still I suggest using stress - strain charts instead of force - displacement.
Response 1: Thank you for pointing this out. We replaced force-displacement charts with stress-strain charts. You can see the details in Figures 4 and 6. Some text changes were made in 3.1 and 3.2.
- Scales in figure 8b should be more visible. I suggest putting scales also in macro images.
Response 2: Thank you for pointing this out. We modified the size and colour of the scales to make scales more obvious and added scales in the macro image. You can see the details in Figure 8 (b).